# Nonlocal Neural Networks, Nonlocal Diffusion and Nonlocal Modeling

**Yunzhe Tao**
School of Engineering and Applied Science
Columbia University, USA
y.tao@columbia.edu

**Qi Sun**
BCSRC & USTC
Beijing, China
sunqi@csrc.ac.cn

**Qiang Du**
School of Engineering and Applied Science
Columbia University, USA
qd2125@columbia.edu

**Wei Liu**
Tencent AI Lab
Shenzhen, China
wl2223@columbia.edu

## Abstract

Nonlocal neural networks [25] have been proposed and shown to be effective in several computer vision tasks, where the nonlocal operations can directly capture long-range dependencies in the feature space. In this paper, we study the nature of diffusion and damping effect of nonlocal networks by doing spectrum analysis on the weight matrices of the well-trained networks, and then propose a new formulation of the nonlocal block. The new block not only learns the nonlocal interactions but also has stable dynamics, thus allowing deeper nonlocal structures. Moreover, we interpret our formulation from the general nonlocal modeling perspective, where we make connections between the proposed nonlocal network and other nonlocal models, such as nonlocal diffusion process and Markov jump process.

## 1 Introduction

Deep neural networks, especially convolutional neural networks (CNNs) [15] and recurrent neural networks (RNNs) [8], have been widely used in a variety of subjects [16]. However, traditional neural network blocks aim to learn the feature representations in a local sense. For example, both convolutional and recurrent operations process a local neighborhood (several nearest neighboring neurons) in either space or time. Therefore, the long-range dependencies can only be captured when these operations are applied recursively, while those long-range dependencies are sometimes significant in practical learning problems, such as image or video classification, text summarization, and financial market analysis [1, 6, 22, 27].

To address the above issue, a nonlocal neural network [25] has been proposed recently, which is able to improve the performance on a couple of computer vision tasks. In contrast to convolutional or recurrent blocks, nonlocal operations [25] capture long-range dependencies directly by computing interactions between each pair of positions in the feature space. Generally speaking, nonlocality is ubiquitous in nature, and the nonlocal models and algorithms have been studied in various domains of physical, biological and social sciences [2, 5, 7, 23, 24].

In this work, we aim to study the nature of nonlocal networks, namely, what the nonlocal blocks have exactly learned through training on a real-world task. By doing spectrum analysis on the weight matrices of the stacked nonlocal blocks obtained after training, we can largely quantify the characteristics of the damping effect of nonlocal blocks in a certain network.

Based on the nature of diffusion observed from experiments, we then propose a new nonlocal neural network which, motivated by the previous nonlocal modeling works, can be shown to be more generic and stable. Mathematically, we can make connections of the nonlocal network to a couple of existing nonlocal models, such as nonlocal diffusion process and Markov jump process. The proposed nonlocal network allows a deeper nonlocal structure, while keeping the long-range dependencies learned in a well-preserved feature space.

## 2 Background

Nonlocal neural networks are usually employed and incorporated into existing cutting-edge model architectures, such as the residual network (ResNet) and its variants, so that one can take full advantage of nonlocal blocks in capturing long-range features. In this section, we briefly review nonlocal networks in the context of traditional image classification tasks and make a comparison among different neural networks. Note that while nonlocal operations are applicable to both space and time variables, we concentrate merely on the spatial nonlocality in this paper for brevity.

### 2.1 Nonlocal Networks

Denote by $X = [X_1, X_2, \cdots, X_M]$ an input sample or the feature representation of a sample, with $X_i$ $(i = 1, \cdots, M)$ being the feature at position $i$. A nonlocal block [25] is defined as

$$Z_i = X_i + \frac{W_Z}{\mathcal{C}_i(X)} \sum_{\forall j} \omega(X_i, X_j) g(X_j) \,. \tag{1}$$

Here $1 \leq i \leq M$, $W_Z$ is the weight matrix, and $Z_i$ is the output signal at position $i$. Computing $Z_i$ depends on the features at possibly all positions $j$. Function $g$ takes any feature $X_j$ as input and returns an embedded representation. The summation is normalized by a factor $\mathcal{C}_i(X)$. Moreover, a pairwise function $\omega$ computes a scalar between the feature at position $i$ and those at all possible positions $j$, which usually represents the similarity or affinity. For simplicity, we only consider $g$ in the form of a linear embedding: $g(X_j) = W_g X_j$, with some weight matrix $W_g$. As is suggested in [25], the choices for the pairwise affinity function $\omega$ can be, but not limited to, (embedded) Gaussian, dot product, and concatenation.

When incorporating nonlocal blocks into a ResNet, a nonlocal network can be written as

$$Z^{k+1} := Z^k + \mathcal{F}(Z^k; W^k) \,, \tag{2}$$

where $W^k$ is the parameter set, $k = 0, 1, \cdots, K$ with $K$ being the total number of network blocks, and $Z^k$ is the output signal at the $k$-th block with $Z^0 = X$, the input sample of the ResNet. On one hand, if a nonlocal block is employed, the $i$-th component of $\mathcal{F}$ will be

$$\left[\mathcal{F}(Z^k; W^k)\right]_i = \frac{W_Z^k}{\mathcal{C}_i(Z^k)} \sum_{\forall j} \omega(Z_i^k, Z_j^k)(W_g^k Z_j^k) \,, \tag{3}$$

where $1 \leq i \leq M$ and $W^k = \{W_Z^k, W_g^k\}$ includes the weight matrices to be learned. In addition, the normalization scalar $\mathcal{C}_i(Z^k)$ is defined as

$$\mathcal{C}_i(Z^k) = \sum_{\forall j} \omega(Z_i^k, Z_j^k) \,, \quad \text{for } i = 1, \cdots, M \,. \tag{4}$$

On the other hand, when the $k$-th block is a traditional residual block of, *e.g.*, the pre-activation ResNet [11], it contains two stacks of batch normalization (BN) [12], rectified linear unit (ReLU) [21], and weight layers, namely,

$$\mathcal{F}(Z^k; W^k) = W_2^k f \left(W_1^k f(Z^k)\right) \,, \tag{5}$$

where $W^k = \{W_1^k, W_2^k\}$ contains the weight matrices, and $f = \text{ReLU} \circ \text{BN}$ denotes the composition of BN and ReLU.

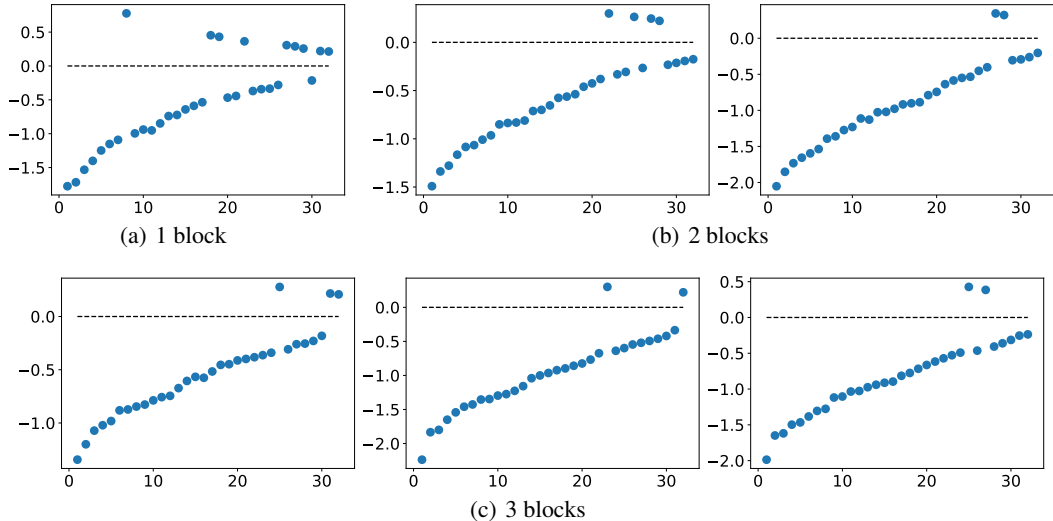

Figure 1: Top 32 eigenvalues of the symmetric part of weight matrices with respect to nonlocal blocks. Figures (a), (b) and (c) correspond to the cases of adding 1, 2 and 3 nonlocal blocks to PreResNet-20, respectively. Y-axis represents the values.

## 2.2 A Brief Comparison with Existing Neural Networks

The most remarkable nature of nonlocal networks is that the nonlocal network can learn long-range correlations since the summation within a nonlocal block is taken over all possible candidates in the current feature space. The pairwise affinity function $\omega$ plays an important role in defining nonlocal blocks, which in some sense determines the level of nonlocality. On one hand, if $\omega$ is always positive, then the feature space will be totally connected and every two features can interact with each other. On the other hand, if $\omega$ is chosen to be a Dirac-delta function $\delta_{ij}$, namely $\omega(X_i, X_j) = 1$ for $i = j$ and is $0$ otherwise, then the nonlocal block gets localized and will act like a simplified residual block.

Besides, as also mentioned in [25], the nature of nonlocal networks is different from other popular neural network architectures, such as CNN [15] and RNN [8]. The convolutional or recurrent operation usually takes the weighted sum over only nearest few neighboring inputs or the latest few time steps, which is still of a local sense in contrast with the nonlocal operation.

## 3 The Damping Effect of Nonlocal Networks

In this section, we first demonstrate the damping effect of nonlocal networks by presenting weight analysis on the well-trained network. More specifically, we train the nonlocal networks for image classification on the CIFAR-10 dataset [14], which consists of 50k training images from 10 classes, and do the spectrum analysis on the weight matrices of nonlocal blocks after training. Based on the analysis, we then propose a more suitable formulation of the nonlocal blocks, followed by some experimental evaluations to demonstrate the effectiveness of the proposed model.

### 3.1 Spectrum Analysis

We incorporate nonlocal blocks into the 20-layer pre-activation ResNet (PreResNet) [11] as stated in Eq. (2). Since our goal is to illustrate the diffusive nature of nonlocal operations, we add a different number of nonlocal blocks into a fixed place at the early stage, which is different from the experiments shown in [25], where the nonlocal blocks are added to different places along the ResNet. When employing the nonlocal blocks, we always insert them to right after the second residual block of PreResNet-20.

The inputs from CIFAR-10 are images of size $32 \times 32$, with preprocessing of the per-pixel mean subtracted. In order to adequately train the nonlocal network, the training starts with a learning rate

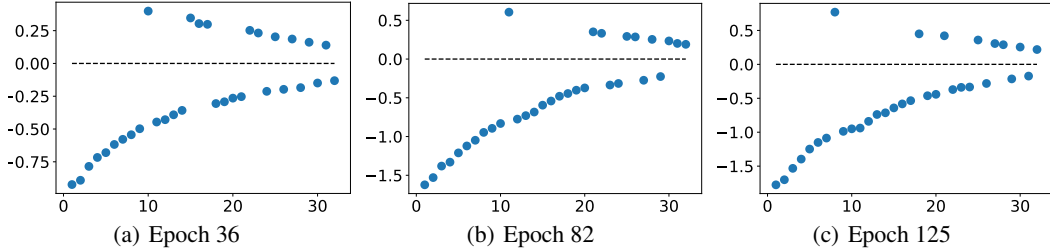

(a) Epoch 36          (b) Epoch 82          (c) Epoch 125

Figure 2: Top 32 eigenvalues of the symmetrized weight matrices during training when adding 1 nonlocal block. Figures (a), (b) and (c) correspond to the results at Epoch 36, 82 and 125, respectively.

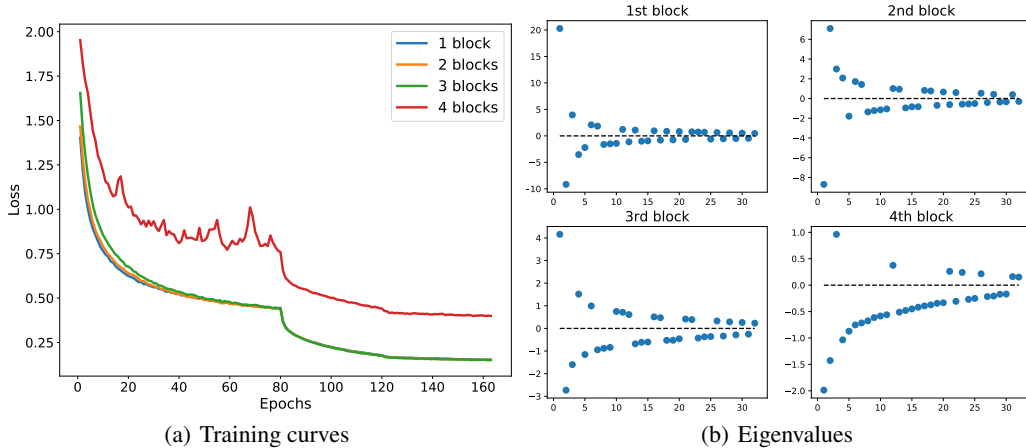

(a) Training curves          (b) Eigenvalues

Figure 3: (a) The training curves of nonlocal networks with 1, 2, 3 and 4 nonlocal blocks. (b) Top 32 eigenvalues of the symmetric part of weight matrices when adding 4 nonlocal blocks.

of $0.1$ that is subsequently divided by 10 at 81 and 122 epochs (around 32k and 48k iterations). A weight decay of $0.0001$ and momentum of $0.9$ are also used. We terminate the training at 164 epochs. The model is with data augmentation [17] and trained on a single NVIDIA Tesla GPU.

To see what the nonlocal blocks have exactly learned, we extract the weight matrices $W_Z$ and $W_g$ after training. Under the current experimental setting, the weight matrices are of dimensions $W_Z \in \mathbb{R}^{64 \times 32}$ and $W_g \in \mathbb{R}^{32 \times 64}$. Note that if we let $W = W_Z W_g$, Eq. (3) can be rewritten as

$$\left[ \mathcal{F}(Z^k \, ; W^k) \right]_i = \frac{W^k}{\mathcal{C}_i(Z^k)} \sum_{\forall j} \omega(Z_i^k, Z_j^k) Z_j^k \, , \tag{6}$$

where the sparse weight matrix $W^k \in \mathbb{R}^{64 \times 64}$ has only 32 eigenvalues. We then denote the symmetric part of $W$ by

$$\widetilde{W} = \frac{W + W^\mathrm{T}}{2} \, , \tag{7}$$

so that the eigenvalues of $\widetilde{W}^k$ are all real. Note that the effect of $W$ on the decay properties of the network is determined by the associated quadratic form, which is equivalent to that of the symmetric part of $W$, *i.e.*, $\widetilde{W}$. Therefore, the eigenvalues of $\widetilde{W}^k$, especially those with the greatest magnitudes (absolute values), would describe the characteristics of the weights in nonlocal blocks.

Figure 1 shows the top 32 eigenvalues of the symmetric part of weight matrices after training when adding 1, 2 and 3 nonlocal blocks to PreResNet-20. One can observe that most of the eigenvalues are negative, especially the first several with the greatest magnitudes. Similar observations can be made along the training process. Figure 2 plots the top 32 eigenvalues of $\widetilde{W}$ at three intermediate epochs before convergence for the 1-block nonlocal network case. From Eq. (2) with $\mathcal{F}$ being the formulation of nonlocal blocks in Eq. (6), we can see that $Z^k$ tends to vanish regardless of the initial

value when imposing small negative coefficients on multiple blocks. Namely, while nonlocal blocks are trying to capture the long-range correlations, the features tend to be damped out at the same time, which is usually the consequence of diffusion. A more detailed discussion can be found in Section 4.4.

However, the training becomes more difficult when employing more nonlocal blocks. Under the current training strategy, when adding 4 blocks, it does not converge after 164 epochs. Although reducing the learning rate or increasing the learning epochs will mitigate the convergence issue, the training loss decreases more slowly than the fewer blocks cases. Figure 3(a) shows the learning curves of nonlocal networks with different nonlocal blocks, where we can see that the training loss for the 4-block network is much larger than the others. Note that for the 4-block case, we have reduced the learning rate by the ratio of 0.1 in order to make the training convergent. This observation implies that the original nonlocal network is not robust with respect to multiple nonlocal blocks, which is also shown in Figure 3(b). For the 4-block case, we obtain many positive eigenvalues of the symmetrized weight matrices after training. In particular, some of the positive eigenvalues have large magnitudes, which leads to a potential blow-up of the feature vectors and thus makes training more difficult.

## 3.2 A New Nonlocal Network

Given the damping effect for the nonlocal network and the instability of employing multiple nonlocal blocks, we hereby suggest to modify the formulation in order to have a more well-defined nonlocal operation. Instead of Eq. (3) or Eq. (6), we regard the nonlocal blocks added consecutively to the same place as a *nonlocal stage*, which consists of several nonlocal sub-blocks. The nonlocal stage takes input $X = [X_i]$ $(i = 1, 2, \cdots, M)$ as the feature representation when computing the affinity within the stage. In particular, a nonlocal stage is defined as

$$Z_i^{n+1} := Z_i^n + \frac{W^n}{\mathcal{C}_i(X)} \sum_{\forall j} \omega(X_i, X_j)(Z_j^n - Z_i^n),$$ (8)

for $i = 1, 2, \cdots, M$, where $W^n$ is the weight matrix to be learned, $Z^0 = X$, and $n = 1, 2, \cdots, N$ with $N$ being the number of stacking sub-nonlocal blocks in a stage. Moreover, $\mathcal{C}_i(X)$ is the normalization factor computed by

$$\mathcal{C}_i(X) = \sum_{\forall j} \omega(X_i, X_j).$$ (9)

There are a couple of differences between the two nonlocal formulations in Eq. (1) and Eq. (8). On one hand, in a nonlocal stage in Eq. (8), the affinity $\omega$ is pre-computed given the input feature $X$ and stays the same along the propagation within a stage, which reduces the computational cost while $\omega(X_i, X_j)$ can still represent the affinity between $Z_i^n$ and $Z_j^n$ to some extent. On the other hand, the residual part of Eq. (8) is no longer a weighted sum of the neighboring features, but the difference between the neighboring signals and computed signal. In Section 4, we will study the connections between the proposed nonlocal blocks and several other existing nonlocal models to clarify the rationale of our model.

## 3.3 Experimental Evaluation

To demonstrate the difference between two nonlocal networks and the effectiveness of our proposed method, we present the empirical evaluation on CIFAR-10 and CIFAR-100. Following the standard practice, we present experiments performed on the training set and evaluated on the test set as validation. We compare the empirical performance of PreResNets incorporating into the original nonlocal blocks [25] or the proposed nonlocal blocks in Eq. (8).

Table 1 presents the validation errors of PreResNet-20 on CIFAR-10 with a different number of nonlocal blocks that are added consecutively to the same place or separately to different places as the experiments shown in [25]. The best performance for each case is displayed in boldface. Note that all the models are trained by ourselves in order to have a fair comparison. We run each model 5 times and report the median. More experimental results with PreResNet-56 on CIFAR-100 are also provided in Table 2. Based on the results shown in the tables, we give some analysis as follows.

First, for the original nonlocal network, since the damping effect cannot be preserved when adding more nonlocal blocks, the training becomes more difficult and thus the validation performs worse. In

Table 1: Validation errors of different models based on PreResNet-20 over CIFAR-10.

| | Model | Error (%) |
|---|---|---|
| | baseline | 8.19 |
| | 2-block (original) | 7.83 |
| | 3-block (original) | 8.28 |
| | 4-block (original) | 15.02 |
| The Same Place | 2-block (proposed) | 7.74 |
| | 3-block (proposed) | 7.62 |
| | 4-block (proposed) | 7.37 |
| | 5-block (proposed) | **7.29** |
| | 6-block (proposed) | 7.55 |
| Different Places | 3-block (original) | 8.07 |
| | 3-block (proposed) | **7.33** |

Table 2: Validation errors of different models based on PreResNet-56 over CIFAR-100.

| | Model | Error (%) |
|---|---|---|
| | baseline | 26.57 |
| | 2-block (original) | 26.13 |
| | 3-block (original) | 26.26 |
| | 4-block (original) | 34.89 |
| PreResNet-56 | 2-block (proposed) | 26.04 |
| | 3-block (proposed) | 25.57 |
| | 4-block (proposed) | 25.43 |
| | 5-block (proposed) | **25.29** |
| | 6-block (proposed) | 25.49 |

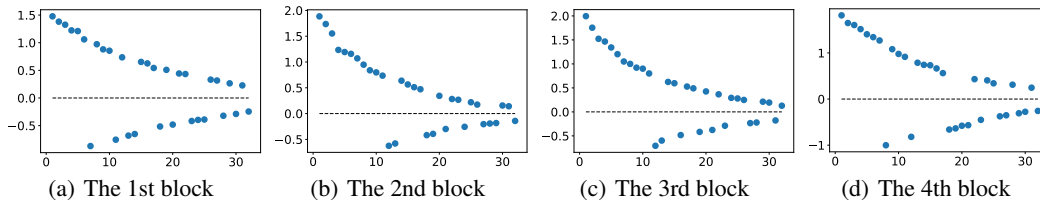

(a) The 1st block     (b) The 2nd block     (c) The 3rd block     (d) The 4th block

Figure 4: Top 32 eigenvalues of the symmetric part of weight matrices after training when adding 4 proposed nonlocal blocks to PreResNet-20.

contrast, the proposed nonlocal network is more robust to the number of blocks. When employing the same number of nonlocal blocks, the proposed model consistently performs better than the original one. Moreover, the proposed model also reduces the computational cost because the affinity function can be pre-assigned in a nonlocal stage. Although the two formulations of nonlocal blocks appear similar, they are different in essence. Figure 4 shows the top 32 eigenvalues of the symmetric part of weight matrices after training when adding 4 new nonlocal blocks to PreResNet-20, where we can see that most of the eigenvalues are positive. In the following section, we will show that the formulation of new nonlocal blocks is exactly the nonlocal analogue of local diffusion with positive coefficients.

In addition, one can observe from Table 1 and Table 2 that adding too many proposed nonlocal blocks reduces the performance. A potential reason is that all the nonlocal blocks in a stage share the same affinity function, which is pre-computed given the input features. Therefore, the kernel function cannot precisely describe the affinity between pairs of features after many sub-blocks.

Note that in [25], the nonlocal blocks are always added individually in each fixed place of ResNets. We also compare the two nonlocal networks in the second part of Table 1, where we in total add 3 nonlocal blocks to PreResNet-20 but into different residual blocks. Although in this case the two kinds of nonlocal blocks have the same affinity kernel, the proposed nonlocal network still performs

better than the original one, which implies that the proposed nonlocal blocks are more suitably defined with respect to the nature of diffusion.

## 4 Connection to Nonlocal Modeling

In this section, we study the relationship between the proposed nonlocal network and a couple of existing nonlocal models, *i.e.*, the nonlocal diffusion systems and Markov chain with jump processes. Then we make a further comparison between the two formulations of nonlocal blocks.

### 4.1 Affinity Kernels

We make some assumptions here on the affinity kernel $\omega$ in order to make connections to other nonlocal modeling approaches. Let $\Gamma$ be the input feature space such that $X_i$'s are finite data samples drawn from $\Gamma$. Denote by $\mathcal{K}$ the kernel matrix such that $\mathcal{K}_{ij} = \omega(X_i, X_j)$. We first assume that $\mathcal{K}$ has a finite Frobenius norm, namely,

$$\|\mathcal{K}\|_F^2 = \sum_{\forall i,j} \left|\omega(X_i, X_j)\right|^2 < \infty\,. \tag{10}$$

Then without loss of generality, we can assume that the kernel matrix $\mathcal{K}$ has sum 1 along its rows, *i.e.*, the normalization factor is assumed to be 1 for all $i$:

$$\mathcal{C}_i(X) = \sum_{\forall j} \omega(X_i, X_j) = 1\,, \quad \text{for } i = 1, \cdots, M\,. \tag{11}$$

We further assume that the kernel function is symmetric and nonnegative within $\Gamma$, namely for all $X_i, X_j \in \Gamma$,

$$\omega(X_i, X_j) = \omega(X_j, X_i) \quad \text{and} \quad \omega(X_i, X_j) \geq 0\,. \tag{12}$$

Note that for the instantiations given in [25], only the Gaussian function is symmetric. However, in the embedded Gaussian and dot product cases, we can instead embed the input features by some weight matrix as the parameter and then feed the pre-embedded features into the nonlocal stage and use the traditional Gaussian or dot product function as the affinity, namely, we replace the kernel by

$$\omega_\theta(X_i, X_j) = \omega\big(\theta(X_i), \theta(X_j)\big)\,, \tag{13}$$

where $\theta$ is a linear embedding function.

### 4.2 Nonlocal Diffusion Process

Define the following discrete nonlocal operator for $X = [X_1, \cdots, X_M]$ and $X_i \in \Gamma$:

$$(\mathcal{L}^h Z)_i := \sum_{\forall j} \omega(X_i, X_j)(Z_j - Z_i)\,, \tag{14}$$

where the superscript $h$ is the discretization parameter. Eq. (14) can be seen as a reformulation of nonlocal operators in some previous works, such as the graph Laplacian [4] (as well as its applications [18–20, 28]), nonlocal-type image processing algorithms [3, 10], and diffusion maps [5]. All of them are nonlocal analogues [7] of local diffusions. Then the equation

$$Z^{n+1} = Z^n + \mathcal{L}^h Z^n \tag{15}$$

describes a discrete nonlocal diffusion, where $Z$ satisfies that $Z^0 = X$ for $X_i \in \Gamma$. The above equation is equivalent to the proposed nonlocal network in Eq. (8) with positive weights.

In general, Eq. (15) is a time-and-space discrete form, where the superscript $n$ is the time step parameter, of the following nonlocal integro-differential equation:

$$\begin{cases} \boldsymbol{z}_t(\boldsymbol{x}, t) - \mathcal{L}\boldsymbol{z}(\boldsymbol{x}) = 0\,, \\ \boldsymbol{z}(\boldsymbol{x}, 0) = \boldsymbol{u}(\boldsymbol{x})\,, \end{cases} \tag{16}$$

for $\boldsymbol{x} \in \Omega$ and $t \geq 0$. Here $\boldsymbol{u}$ is the initial condition, $\boldsymbol{z}_t := \partial \boldsymbol{z}/\partial t$, and $\mathcal{L}$ defines a continuum nonlocal operator:

$$\mathcal{L}\boldsymbol{z}(\boldsymbol{x}) := \int_\Omega \rho(\boldsymbol{x}, \boldsymbol{y})\big(\boldsymbol{z}(\boldsymbol{y}) - \boldsymbol{z}(\boldsymbol{x})\big)d\boldsymbol{y}\,, \tag{17}$$

where the kernel

$$\rho(\boldsymbol{x}, \boldsymbol{y}) := \omega\big(\boldsymbol{u}(\boldsymbol{x}), \boldsymbol{u}(\boldsymbol{y})\big) \qquad (18)$$

is symmetric and positivity-preserved. We provide in the supplementary material some properties of the nonlocal equation (16), in order to illustrate the connection between Eq. (16) and local diffusions, and demonstrate that the proposed nonlocal blocks can be viewed as an analogue of local diffusive terms.

Since we assume that the kernel matrix $\mathcal{K}$ has a finite Frobenius norm, we have

$$\int_\Omega \int_\Omega |\rho(\boldsymbol{x}, \boldsymbol{y})|^2 d\boldsymbol{x} d\boldsymbol{y} < \infty. \qquad (19)$$

Therefore, the corresponding integral operator $\mathcal{L}$ is a Hilbert-Schmidt operator so that Eq. (16) and its time reversal are both well-posed and stable in finite time. The latter is equivalent to the continuum generalization of the nonlocal block with a negative coefficient (eigenvalue). Although $\mathcal{L}$ is a nonlocal analogue of the local Laplace (diffusion) operator, the nonlocal diffusion equation has its own merit. More specifically, the anti-diffusion in a local partial differential equation (PDE), *i.e.*, the reversed heat equation, is ill-posed and unstable in finite time. The stability property of the proposed nonlocal network is important, because it ensures that we can stack multiple nonlocal blocks within a single stage to fully exploit their advantages in capturing long-range features.

### 4.3 Markov Jump Process

The proposed nonlocal network in Eq. (8) also shares some common features with the discrete-time Markov chain with jump processes [7]. In this part, we assume, without loss of generality, that $Z_i$'s and $X_i$'s are scalars. In general, the component-wise properties can lead to similar properties of the vector field. Given a Markov jump process $\mathcal{Z}_t$ confined to remain in a bounded domain $\Omega \subset \mathbb{R}^d$, assume that $z(\boldsymbol{x}, t)$ is the corresponding probability density function. Then a general master equation to describe the evolution of $z$ [13] can be written as

$$z_t(\boldsymbol{x}, t) = \int_\Omega \left[ \gamma(\boldsymbol{x}', \boldsymbol{x}, t) z(\boldsymbol{x}', t) - \gamma(\boldsymbol{x}, \boldsymbol{x}', t) z(\boldsymbol{x}, t) \right] d\boldsymbol{x}', \qquad (20)$$

where $\gamma(\boldsymbol{x}', \boldsymbol{x}, t)$ denotes the transition rate from $\boldsymbol{x}'$ to $\boldsymbol{x}$ at time $t$. Assume that the Markov process is time-homogeneous, namely $\gamma(\boldsymbol{x}', \boldsymbol{x}, t) = \gamma(\boldsymbol{x}', \boldsymbol{x})$. Then in the discrete time form [29], Eq. (20) is often reformulated as

$$z(\boldsymbol{x}, t) - z(\boldsymbol{x}, t') = \int_\Omega \left[ p(\boldsymbol{x}', \boldsymbol{x}) z(\boldsymbol{x}', t') - p(\boldsymbol{x}, \boldsymbol{x}') z(\boldsymbol{x}, t') \right] d\boldsymbol{x}', \qquad (21)$$

where $p(\boldsymbol{x}', \boldsymbol{x}) := (t - t') \gamma(\boldsymbol{x}', \boldsymbol{x})$ represents the transition probability of a particle moving from $\boldsymbol{x}'$ to $\boldsymbol{x}$. One can easily see that as $t' \to t$, the solution to Eq. (21) converges to the continuum solution to Eq. (20). Note that $\int_\Omega p(\boldsymbol{x}, \boldsymbol{x}') d\boldsymbol{x}' = 1$, Eq. (21) reduces to

$$z(\boldsymbol{x}, t) = \int_\Omega p(\boldsymbol{x}', \boldsymbol{x}) z(\boldsymbol{x}', t') d\boldsymbol{x}'. \qquad (22)$$

Suppose that there is a set of finite states drawn from $\Omega$, namely $\boldsymbol{x}_1, \boldsymbol{x}_2, \cdots, \boldsymbol{x}_M$, and finite discrete time intervals, namely $t_1, t_2, \cdots, t_N$. Let $Z_i^n = z(\boldsymbol{x}_i, t_n)$ with an initial condition $Z_i^0 = X_i$. If we further introduce the kernel matrix $\mathcal{K}$, where $\mathcal{K}_{ij} := \frac{1}{\mathcal{M}(\boldsymbol{x}'_j)} \int_{\mathcal{M}(\boldsymbol{x}'_j)} p(\boldsymbol{x}_i, \boldsymbol{x}') d\boldsymbol{x}'$, and the average of $p(\boldsymbol{x}_i, \boldsymbol{x}')$ over the grid mesh of $\boldsymbol{x}'_j$. Then the kernel matrix $\mathcal{K}$ is a *Markov matrix* (with each column summing up to 1) and

$$Z_i^{n+1} = \sum_{\forall j} \mathcal{K}_{ij} Z_j^n \qquad (23)$$

describes a discrete-time Markov jump process. The above system is equivalent to

$$Z_i^{n+1} - Z_i^n = \sum_{\forall j} \mathcal{K}_{ij} \left( Z_j^n - Z_i^n \right), \qquad (24)$$

which is exactly the proposed nonlocal stage regardless of the weight coefficients in front of the Markov operator.

On the other hand, our proposed nonlocal network with negative weight matrices can be regraded as the reverse Markov jump process. Again, since we have the condition that

$$\|\mathcal{K}\|_F^2 < \infty \,, \tag{25}$$

the Markov operator $\mathcal{K}$ is a Hilbert-Schmidt operator, which is bounded and implies that the Markov jump processes with both time directions are stable in finite time.

## 4.4 Further Discussion

In this section, we have derived some properties of the proposed nonlocal network. Due to the nature of a Hilbert-Schmidt operator, we expect the robustness of our network. However, we should remark that whether the exact discrete formula in Eq. (8) has stable dynamics also depends on the weights $W^n$'s. This is ignored for simplicity when connecting to other nonlocal models. In practice, the stability holds as long as the weight parameters are small enough such that the *CFL condition* is satisfied.

In comparison, for the original nonlocal network, we define a discrete nonlinear nonlocal operator as

$$(\tilde{\mathcal{L}}^h Z)_i := -\sum_{\forall j} \omega(Z_i, Z_j) Z_j \,. \tag{26}$$

Then the flow of the original nonlocal network with small negative coefficients can be described as

$$Z^{n+1} - Z^n = \tilde{\mathcal{L}}^h Z^n \,, \tag{27}$$

with the initial condition $Z^0 = X$. By letting $Z^{n+1} = Z^n$, the steady-state equation of Eq. (27) can be written as

$$\tilde{\mathcal{L}}^h Z = 0 \,. \tag{28}$$

Since the kernel $\omega$ is strictly non-negative, the only steady-state solution is $Z \equiv 0$, which means that the output signals of original nonlocal blocks tend to be damped out (diffused) along iterations. However, the original nonlocal operation with positive eigenvalues is unstable in finite time. While one can still learn long-range features in the network, stacking several original blocks will cast uncertainty to the model and thus requires extra work to study the initialization strategy or to fine-tune the parameters.

In applications, the nonlocal stage is usually employed and plugged into ResNets. From the viewpoint of PDEs, the flow of a ResNet can be seen as the forward Euler discretization of a dynamical system with respect to $\boldsymbol{z}(t)$ [26]:

$$\frac{d\boldsymbol{z}}{dt} = F(\boldsymbol{z}, W(t)), \quad \boldsymbol{z}(0) = X \,. \tag{29}$$

The right-hand side $F$ for a PreResNet is given by

$$F(\boldsymbol{z}, W(t)) = W_2(t) f\left(W_1(t) f(\boldsymbol{z})\right) \,, \tag{30}$$

which can be regarded as a reaction term in a PDE because it does not involve any differential operator explicitly. From the standard PDE theory, incorporating the proposed nonlocal stage to ResNets is equivalent to introducing diffusion terms to the reaction system. As a result, the diffusion terms can regularize the PDE and thus make it more stable [9].

## 5 Conclusion

In this paper, we studied the damping effect of the existing nonlocal networks in the literature by performing the spectrum analysis on the weight matrices of the well-trained networks. Then we proposed a new class of nonlocal networks, which can not only capture the long-range dependencies but also be shown to be more stable in the network dynamics and more robust to the number of nonlocal blocks. Therefore, we can stack more nonlocal blocks in order to fully exploit their advantages. In the future, we aim to investigate the proposed nonlocal network on other challenging real-world learning tasks.

## Acknowledgments

This work is supported in part by US NSF CCF-1740833, DMR-1534910 and DMS-1719699. Y. Tao wants to thank Tingkai Liu and Xinpeng Chen for their help on the experiments.

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
