[Supplementary Material · supplementary_NIPS.pdf]

# Supplementary Material to Nonlocal Neural Networks, Nonlocal Diffusion and Nonlocal Modeling

In the supplementary material, we aim to show some decay properties of the nonlocal equation

$$\begin{cases} z_t(\boldsymbol{x}, t) - \mathcal{L}z(\boldsymbol{x}) = 0 \,, \\ z(\boldsymbol{x}, 0) = u(\boldsymbol{x}) \,, \end{cases} \tag{1}$$

for $\boldsymbol{x} \in \Omega \subset \mathbb{R}^d$ and $t \geq 0$. Here the nonlocal operator is defined as

$$\mathcal{L}z(\boldsymbol{x}) := \int_\Omega \rho(\boldsymbol{x}, \boldsymbol{y})\big(z(\boldsymbol{y}) - z(\boldsymbol{x})\big)d\boldsymbol{y} \,, \tag{2}$$

where the kernel

$$\rho(\boldsymbol{x}, \boldsymbol{y}) := \omega(u(\boldsymbol{x}), u(\boldsymbol{y})) \tag{3}$$

is symmetric and positivity-preserved. We should remark first that in all the proofs presented below, we always omit the time variable $t$ if not referred to. For simplicity, we always assume that $z(\boldsymbol{x})$ is a scalar function. We first derive some properties of the nonlocal operator $\mathcal{L}$.

**Proposition 1.** *The nonlocal operator $\mathcal{L}$ defined in Eq. (2) admits the following properties:*

(i) *If $z(\boldsymbol{x})$ is a constant function for all $\boldsymbol{x} \in \Omega$, then $\mathcal{L}z \equiv 0$.*
(ii) *$\mathcal{L}z$ is mean zero for any function $z$, namely,*

$$\int_\Omega \mathcal{L}z(\boldsymbol{x})d\boldsymbol{x} = 0 \,. \tag{4}$$

(iii) *$-\mathcal{L}$ is positive semi-definite, namely,*

$$-\int_\Omega \mathcal{L}z(\boldsymbol{x})z(\boldsymbol{x})d\boldsymbol{x} \geq 0 \,, \tag{5}$$

*for any function $z$.*

*Proof.* Property (i) is straightforward. Property (ii) is true by the symmetry of the kernel function, namely,

$$\begin{aligned} \int_\Omega \mathcal{L}z(\boldsymbol{x})d\boldsymbol{x} &= \int_\Omega \int_\Omega \rho(\boldsymbol{x}, \boldsymbol{y})(z(\boldsymbol{y}) - z(\boldsymbol{x}))d\boldsymbol{y}d\boldsymbol{x} \\ &= \frac{1}{2}\int_\Omega \int_\Omega \rho(\boldsymbol{x}, \boldsymbol{y})(z(\boldsymbol{y}) - z(\boldsymbol{x}))d\boldsymbol{y}d\boldsymbol{x} + \frac{1}{2}\int_\Omega \int_\Omega \rho(\boldsymbol{y}, \boldsymbol{x})(z(\boldsymbol{x}) - z(\boldsymbol{y}))d\boldsymbol{y}d\boldsymbol{x} \\ &= 0 \,. \end{aligned}$$

For Property (iii), using the symmetry of $\rho$ again we can get

$$\begin{aligned} -\int_\Omega \mathcal{L}z(\boldsymbol{x})z(\boldsymbol{x})d\boldsymbol{x} &= -\int_\Omega \int_\Omega \rho(\boldsymbol{x}, \boldsymbol{y})(z(\boldsymbol{y}) - z(\boldsymbol{x}))z(\boldsymbol{x})d\boldsymbol{y}d\boldsymbol{x} \\ &= \frac{1}{2}\int_\Omega \int_\Omega \rho(\boldsymbol{x}, \boldsymbol{y})(\boldsymbol{z}(\boldsymbol{y}) - \boldsymbol{z}(\boldsymbol{x}))^2 d\boldsymbol{y}d\boldsymbol{x} \,. \end{aligned}$$

Since $\rho(\boldsymbol{x}, \boldsymbol{y}) \geq 0$ for all $\boldsymbol{x}, \boldsymbol{y} \in \Omega$, the above is always non-negative. $\qquad\square$

It is easy to see that the discrete nonlocal operator $\mathcal{L}^h$ defined in Section 4.2 of the main paper has the same properties in the discrete form. Now we can study the properties of the nonlocal integral-differential equation (1). The following gives a formal theorem on this observation.

**Theorem 2.** *The flow of Eq.* (1) *admits the following properties:*

  (i) *The mean value is preserved, namely,*

$$\int_\Omega z(t,\boldsymbol{x})d\boldsymbol{x} = \int_\Omega u(\boldsymbol{x})d\boldsymbol{x}, \quad \text{for all } t \geq 0\,. \tag{6}$$

 (ii) *As $t \to \infty$, the solution converges to a constant, which is the mean value of the initial condition u, namely,*

$$\lim_{t\to\infty} z(t,\boldsymbol{x}) = \frac{1}{|\Omega|} \int_\Omega u(\boldsymbol{x})d\boldsymbol{x}\,, \tag{7}$$

*where $|\Omega|$ is the measure of the space domain.*

*Proof.* For Property (i), by taking the time derivative of the integral of $z$ over $\Omega$, we can get that

$$\frac{d}{dt}\int_\Omega z(t,\boldsymbol{x})d\boldsymbol{x} = \int_\Omega z_t(t,\boldsymbol{x})d\boldsymbol{x} = \int_\Omega \mathcal{L}z(\boldsymbol{x})d\boldsymbol{x}\,.$$

From Property (ii) of Proposition 1, the above equations are equal to 0, which means that the mean value does not change over time.

For Property (ii), we first prove another property of energy decay, namely

$$\frac{d}{dt}\int_\Omega z(t,\boldsymbol{x})^2 d\boldsymbol{x} \leq 0\,.$$

Indeed, by Property (iii) of Proposition 1, we have

$$\frac{d}{dt}\int_\Omega z(t,\boldsymbol{x})^2 d\boldsymbol{x} = 2\int_\Omega z_t(t,\boldsymbol{x})z(t,\boldsymbol{x})d\boldsymbol{x}$$
$$= 2\int_\Omega \mathcal{L}z(\boldsymbol{x})z(\boldsymbol{x})d\boldsymbol{x}$$
$$\leq 0\,.$$

Moreover, in the proof of Property (iii) of Proposition 1, we can see that the above estimate is strictly negative unless $z(\boldsymbol{x})$ is a constant for any $\boldsymbol{x} \in \Omega$. Then as $t \to \infty$, $z(t,\boldsymbol{x})$ must converge to a constant function, which has to be $\frac{1}{|\Omega|}\int_\Omega u(\boldsymbol{x})d\boldsymbol{x}$ by Property (i) of this theorem. $\qquad\square$

Another observation is that the variance of $z(t,\boldsymbol{x})$ is decreasing over iterations. The variance is defined as

$$\text{var}(z) := \frac{1}{|\Omega|}\int_\Omega \left(z(\boldsymbol{x}) - \frac{1}{|\Omega|}\int_\Omega z(\boldsymbol{y})d\boldsymbol{y}\right)^2 d\boldsymbol{x}\,. \tag{8}$$

Then we can derive the following theorem.

**Theorem 3.** *Let $z(t,\boldsymbol{x})$ be the solution to Eq.* (1)*, then*

$$\frac{d}{dt}var(z) \leq 0\,. \tag{9}$$

*Proof.* Taking the time derivative of the variance, we can get

$$\frac{d}{dt}\text{var}(z) = \frac{2}{|\Omega|}\int_\Omega \left(z(t,\boldsymbol{x}) - \frac{1}{|\Omega|}\int_\Omega u(\boldsymbol{y})d\boldsymbol{y}\right)z_t(t,\boldsymbol{x})d\boldsymbol{x}$$
$$= \frac{2}{|\Omega|}\left(\int_\Omega z(\boldsymbol{x})\mathcal{L}z(\boldsymbol{x})d\boldsymbol{x} - \frac{1}{|\Omega|}\int_\Omega u(\boldsymbol{y})d\boldsymbol{y}\int_\Omega \mathcal{L}z(\boldsymbol{x})d\boldsymbol{x}\right)$$
$$= \frac{2}{|\Omega|}\int_\Omega z(\boldsymbol{x})\mathcal{L}z(\boldsymbol{x})d\boldsymbol{x}$$
$$\leq 0\,,$$

where in the derivation we have used the fact that $\int_\Omega \mathcal{L}z(\boldsymbol{x})d\boldsymbol{x} = 0$. $\qquad\square$

We are also interested in the convergence rate of $z(t, \boldsymbol{x})$ to its mean value with respect to $t$. The following theorem tells us that the solution to Eq. (1) decays to its mean value with an exponential rate.

**Theorem 4.** *Suppose $u \in L^2(\Omega)$, i.e., square integrable. Then as $t \to \infty$, the solution $z$ to Eq. (1) satisfies*

$$\|z(t, \boldsymbol{x}) - \bar{u}\|_{L^2(\Omega)} \leq Ce^{-\lambda t},\tag{10}$$

*with some positive constants $C$ and $\lambda$, where*

$$\bar{u} = \frac{1}{|\Omega|}\int_\Omega u(\boldsymbol{x})d\boldsymbol{x}.\tag{11}$$

Before we give the proof of Theorem 4, we establish a nonlocal-type Poincaré inequality as follows.

**Lemma 5 (Nonlocal Poincaré Inequality).** *Supposed that $z(\boldsymbol{x}) \in L^2(\Omega)$ and $\int_\Omega z(\boldsymbol{x})d\boldsymbol{x} = 0$. Then there exists a constant $C > 0$ such that*

$$\int_\Omega z(\boldsymbol{x})^2 d\boldsymbol{x} \leq C \int_\Omega \int_\Omega \rho(\boldsymbol{x}, \boldsymbol{y})(z(\boldsymbol{y}) - z(\boldsymbol{x}))^2 d\boldsymbol{y}d\boldsymbol{x}.$$

*Proof.* We show that

$$0 < m = \inf_{z(\boldsymbol{x}) \in L^2(\Omega), \|z\|_{L^2} = 1} \left( \int_\Omega \int_\Omega \rho(\boldsymbol{x}, \boldsymbol{y})(z(\boldsymbol{y}) - z(\boldsymbol{x}))^2 d\boldsymbol{y}d\boldsymbol{x} \right).$$

Clearly, $m \geq 0$. Suppose that $m = 0$, then there exists a sequence $z_n(\boldsymbol{x}) \in L^2(\Omega)$ and $\int_\Omega z_n(\boldsymbol{x})d\boldsymbol{x} = 0$ such that, for all $n$, $\|z_n\|_{L^2} = 1$ and

$$\lim_{n \to \infty} \int_\Omega \int_\Omega \rho(\boldsymbol{x}, \boldsymbol{y})(z_n(\boldsymbol{y}) - z_n(\boldsymbol{x}))^2 d\boldsymbol{y}d\boldsymbol{x} = 0.$$

Then $\{z_n\}$ is precompact in $L^2(\Omega)$, which means that there exists a strong limit $z_\infty$ of $z_n$ in $L^2(\Omega)$. Then, on one hand, we have $\|z_\infty\|_{L^2} = 1$ and $\int_\Omega z_\infty(\boldsymbol{x})d\boldsymbol{x} = 0$. On the other hand,

$$\begin{aligned}
0 &= \lim_{n \to \infty} \int_\Omega \int_\Omega \rho(\boldsymbol{x}, \boldsymbol{y})(z_n(\boldsymbol{y}) - z_n(\boldsymbol{x}))^2 d\boldsymbol{y}d\boldsymbol{x} \\
&= \lim_{n \to \infty} -2 \int_\Omega \mathcal{L}z_n(\boldsymbol{x})z_n(\boldsymbol{x})d\boldsymbol{x} \\
&= -2 \int_\Omega \mathcal{L}z_\infty(\boldsymbol{x})z_\infty(\boldsymbol{x})d\boldsymbol{x} \\
&= \int_\Omega \int_\Omega \rho(\boldsymbol{x}, \boldsymbol{y})(z_\infty(\boldsymbol{y}) - z_\infty(\boldsymbol{x}))^2 d\boldsymbol{y}d\boldsymbol{x}.
\end{aligned}$$

The above integral is 0 if and only if $z_\infty$ is a constant function. Then by $\int_\Omega z_\infty(\boldsymbol{x})d\boldsymbol{x} = 0$, we must have $z_\infty \equiv 0$, which contradicts to $\|z_\infty\|_{L^2} = 1$. This completes the proof. $\qquad\square$

Now we can start to prove Theorem 4.

*Proof of Theorem 4.* Let $\bar{z}(t, \boldsymbol{x}) = z(t, \boldsymbol{x}) - \bar{u}$. Then we have

$$\int_\Omega \bar{z}(t, \boldsymbol{x})d\boldsymbol{x} = 0 \quad \text{for all } t \geq 0.$$

Since $u \in L^2(\Omega)$, it is also easy to get that $\bar{z}(\cdot, \boldsymbol{x}) \in L^2(\Omega)$. Moreover, $\bar{z}$ is the solution to

$$\begin{cases}
\bar{z}_t(t, \boldsymbol{x}) - \displaystyle\int_\Omega \rho(\boldsymbol{x}, \boldsymbol{y})(\bar{z}(t, \boldsymbol{y}) - \bar{z}(t, \boldsymbol{x}))d\boldsymbol{y} = 0, \\
\bar{z}(0, \boldsymbol{x}) = u(\boldsymbol{x}) - \bar{u}.
\end{cases}$$

Then by the nonlocal Poincaré inequality, there exists a constant $\lambda > 0$ such that

$$\frac{d}{dt}\int_\Omega \bar{z}(t,\boldsymbol{x})^2 d\boldsymbol{x} = -\int_\Omega \rho(\boldsymbol{x},\boldsymbol{y})(\bar{z}(\boldsymbol{y}) - \bar{z}(\boldsymbol{x}))^2 d\boldsymbol{y}d\boldsymbol{x}$$

$$\leq -2\lambda \int_\Omega \bar{z}(\boldsymbol{x})^2 d\boldsymbol{x}\,.$$

Therefore, we can derive that

$$\frac{d}{dt}\left(e^{2\lambda t}\int_\Omega \bar{z}(t,\boldsymbol{x})^2 d\boldsymbol{x}\right) \leq 0\,.$$

Let $C = \left(\int_\Omega \bar{z}(0,\boldsymbol{x})^2 d\boldsymbol{x}\right)^{1/2}$. Then we have

$$\|z(t,\boldsymbol{x}) - \bar{u}\|_{L^2(\Omega)} \leq Ce^{-\lambda t}\,.$$

$\square$

To sum up, all the properties derived from the nonlocal equation are also true for local diffusions. Moreover, since the nonlocal equation (1) is a continuum generalization of the proposed nonlocal network which can be regarded as the discretization of Eq. (1), we can view the formulation of the proposed nonlocal blocks as an analogue of local diffusive terms.