[Reviews · NeurIPS 2018]

Reviewer 1



This paper followed the work of nonlocal neural network to discuss the properties of the diffusion and damping effect by analyzing the spectrum. The trained nonlocal network for image classification on CIFSR-10 data by incorporating nonlocal blocks into the 20-layer PreResNet presented most eigenvalues to be negative and convergence challenges when more blocks were added under certain learning rate and epochs. A rough look at the nonlocal operator representation under steady-state shed light that the output signals of the original nonlocal blocks tend to be damped out (diffused) along iterations by design. A new nonlocal network with namely nonlocal stage component was proposed to help overcome the aforementioned damped out problem by essentially replacing the residual part from weighted sum of the neighboring features to the difference between the neighboring signals and computed signals. Another proposed change is replacing the pairwise affinity function based on updated output to the input feature, which stays the same along the propagation with a stage. The new proposed nonlocal network outperforms the original nonlocal network in the previous mentioned data example in the sense of providing smaller validation errors under various block cases (in the same/different residual blocks). We also see about 2/3 of eigenvalues are positive. Meanwhile, we see the validation error rate first decrease and then increase again when more nonlocal blocks are added. A light touch on the nonlocal diffusion process representation and discrete-time Markov chain with nonlocal jump representation help understand the stability benefit of the proposed nonlocal network. The structure and idea of the paper is clear, would be a plus if the connection to nonlocal modeling part be more rigorously written and a little more introduction/comparison with closely related/cited method. A few minor detailed comments are 1) Full name of the method when first appear like ResNet in Line 38 and ReLU in Line 61. 2) Might help reader to better follow if briefly point out the relation of Z^{k} in (2) and Z_{i} in (1) when first appear, without having to digest based on later (3). 3) Define weight matrix W_{z} in (1), clarify the h of \mathcal{L}^{h} in (14) is discretization parameter, make it clear in (16) z_{t} is \partial{z}/\partial{t}.

Reviewer 2



This paper provided some theoretical analysis for the nonlocal neural network model, by studying the spectral distribution of the weight matrices. Furthermore, the authors proposed a new form of nonlocal formulation, and connected it to standard stochastic processes such as nonlocal diffusion processes and Markov jump processes, to demonstrate the advantage of the new formulation. However, I am not fully convinced of the advantage of the proposed new formulation, for the following reasons: 1) The connection between the proposed nonlocal formulation and the standard stochastic processes are not very strong. For example, equation (14) is a simplified version of the proposed model in equation (8), hence the conclusions about stableness can not be readily applied to equation (8). 2) Although the authors commented in Section 4.4 that "the stability holds as long as the weight parameters are small enough ...", this statement is not rigorous enough to support the stableness claim. 3) Improvement is obtained with the new formulation on CIFAR-10, but relatively small. It would be better to test the proposed model on more datasets, which may give stronger support for the proposed model. Note: I have read the authors' response and agree that "the new one is more robust in stacking multiple nonlocal blocks". In addition, the authors provided more experimental results with CIFAR-100, although the improvement compared to the baseline is still not big. Based on the above information, I changed my score from 5 to 6.

Reviewer 3



This paper studies the damping effect of a non-local neural network (Want et al., CVPR 2018) by performing spectrum analysis on the weight matrices obtained after well training. Given the eigenvalues of the weight matrices, the paper claimed that non-local blocks, different from local learning, have the nature of diffusion. The authors then proposed an alternative formula of the non-local blocks, which is inspired by some previous classical non-local modeling works (Buades et al. 2005, Coifman et al. 2006, Gilboa et al. 2007, Du et al. 2012). A similarity between the proposed non-local operation and other non-local models, i.e., nonlocal diffusion process and Markov jump process, has been shown in the paper to demonstrate the diffusive nature and stability of the proposed non-local blocks. The experimental results also showed that the modified non-local network not only gets higher accuracy on image classification tasks than the original one, but also is more robust to multiple blocks. Strengths: 1) The paper is well written and organized, and technically correct. 2) I appreciate the connections with a wide range of existing non-local models. It is always an enjoy to read papers that try to find theoretical justifications for deep neural architectures. In this sense, the paper is a refreshing read. Although the spectrum analysis technique has been widely studied and applied, using it to interpret deep learning seems useful yet simple. 3) Other than the theoretical findings, another contribution of this paper is that it provides a modified non-local network architecture. Although the two formulations look similar, I am convinced by the justifications in Section 4 of the paper and Section 3 of the supplementary material that the modification makes the whole system more stable and well-defined. 4) The experiments are self-contained and kept to the point. In the original paper of non-local network (Wang et al., CVPR 2018), although it was investigated in many datasets and tasks that introducing non-local blocks can improve the learning performance, the non-local blocks are always added to different places. While in this paper, the authors studied the non-local network from a different perspective, namely stacking multiple blocks to the same place, serving for the study of stability and robustness. Then the bottom results in Table 1 also gave a comparison of the two non-local networks with non-local blocks incorporated into different places. Weaknesses: Although this paper paid more attention to the theoretical justification of the non-local network architecture, and the empirical results are presented to validate the theoretical findings, I am curious about the application of the modified non-local network. Since in (Wang et al., CVPR 2018), the authors provided complete experiments on a few tasks and datasets, especially the video classification on Kinects. I am willing to see how the modified non-local network performs on such a task with multiple non-local blocks. Overall, the tools used in this paper are useful and extendable. The topic raised in the paper is interesting to the community. The proposed model architecture is incremental, but the theoretical study is original and refreshing. The significance of the paper does not lie in the improvement of experimental performance, but a new angle of studying the neural network architecture. Using some mathematical tools, such as PDEs, ODEs, and stochastic processes, to interpret and explore deep learning techniques seems promising. This is a nice paper. The rebuttal makes the research idea clearer. I keep my vote to accept this paper.